# Structure of the human BBSome core complex

**Björn Udo Klink[1], Christos Gatsogiannis[1], Oliver Hofnagel[1], Alfred Wittinghofer[2], Stefan Raunser[1]\***

[1]Department of Structural Biochemistry, Max Planck Institute of Molecular Physiology, Dortmund, Germany; [2]Structural Biology Group, Max Planck Institute of Molecular Physiology, Dortmund, Germany

**Abstract** The BBSome is a heterooctameric protein complex that plays a central role in primary cilia homeostasis. Its malfunction causes the severe ciliopathy Bardet-Biedl syndrome (BBS). The complex acts as a cargo adapter that recognizes signaling proteins such as GPCRs and links them to the intraflagellar transport machinery. The underlying mechanism is poorly understood. Here we present a high-resolution cryo-EM structure of a human heterohexameric core subcomplex of the BBSome. The structure reveals the architecture of the complex in atomic detail. It explains how the subunits interact with each other and how disease-causing mutations hamper this interaction. The complex adopts a conformation that is open for binding to membrane-associated GTPase Arl6 and a large positively charged patch likely strengthens the interaction with the membrane. A prominent negatively charged cleft at the center of the complex is likely involved in binding of positively charged signaling sequences of cargo proteins.

## Introduction

Ciliary research had a rocky trail to travel since the discovery of cilia in 1675 as the first known organelle by Antony Van Leeuwenhoek. Although primary cilia have long been thought to be only minor players in the cellular opera (*Bloodgood, 2009*), they perform key functions as cellular antennae. Primary cilia contain a plethora of crucial signaling proteins with important sensory and regulatory functions particularly in development and cell signaling (*Satir, 2017*; *Nachury, 2014*). Since cilia do not contain a protein synthesis machinery, a key question of ciliary research is how proteins are transported to, from and within the cilium. The BBSome is a cargo adaptor that recognizes a diverse set of membrane-bound ciliary proteins. It binds with its cargo to the intraflagellar transport (IFT) complex, a large heterooligomeric protein complex that is transported along ciliary microtubules by the molecular motors dynein and kinesin (*Wingfield et al., 2018*; *Stepanek and Pigino, 2016*). Interestingly, the BBSome is also involved in the assembly and stabilization of the IFT complex (*Wei et al., 2012*; *Ou et al., 2005*), with varying impact on IFT stability in different organisms (*Wei et al., 2012*; *Williams et al., 2014*; *Pan et al., 2006*; *Lechtreck et al., 2009*).

It has been proposed that the BBSome is assembled in a sequential order starting from a complex of BBS7, BBS chaperonins and the CCT/TRiC complex, which acts as a scaffold for further BBSome subunits to be added (*Zhang et al., 2012a*). However, this route cannot hold true in *Drosophila*, where the BBSome components BBS2 and BBS7 are missing. The lack of these domains in *Drosophila* raises in general the question whether these subunits are important for central BBsome functions such as cargo and membrane binding or if they are mostly involved in cilia-specific processes that require the interaction with the IFT complex. In line with this, *Drosophila* and other organisms with a small number of ciliated cells lack the IFT proteins IFT25 and IFT27 (*Zhang et al., 2017*), which are considered to be the anchor points for the BBSome on the IFT complex (*Liew et al., 2014*; *Eguether et al., 2014*; *Lechtreck, 2015*). It is therefore conceivable that BBS2

**\*For correspondence:**
stefan.raunser@mpi-dortmund.mpg.de

**Competing interests:** The authors declare that no competing interests exist.

and BBS7 perform functions that are mostly relevant in the context of the cilium, while the other BBSome subunits BBS1, 4, 5, 8, 9, and 18 represent a core complex with an independent set of functions, some of which might be relevant outside the cilium.

BBS1 emerged to be the most important BBSome subunit for cargo recognition, with several described interactions with ciliary cargo proteins (*Jin et al., 2010*; *Seo et al., 2011*; *Su et al., 2014*; *Seo et al., 2009*; *Bhogaraju et al., 2013*; *Ruat et al., 2012*; *Zhang et al., 2012b*). The BBSome is recruited to membranes by the small GTPase Arl6 (*Jin et al., 2010*), which binds to the N-terminal β-propeller domain of BBS1. The crystal structures of this domain in complex with Arl6 (*Mourão et al., 2014*), as well as the β-propeller of BBS9 (24), provide the only currently available high-resolution structural information on BBSome subdomains. While this manuscript was in preparation, the medium-resolution structure of a BBSome complex purified from bovine retina was reported (*Chou et al., 2019*). The structure revealed that BBS2 and BBS7 form a top lobe that blocks the Arl6 interaction site on BBS1. The complex, however, has been purified by affinity chromatography using Arl6 as bait. This suggests that the conformation of the BBsome in its apo state differs from that of the complex bound to Arl6.

Although the cryo-EM structure at 4.9 Å revealed the overall domain architecture of the complex, an atomic model could not be accurately built due to the limited resolution. However, this is needed to fully understand the interactions of the domains and the mechanism underlying cargo binding and membrane interaction.

We recently reconstituted a heterologously expressed core complex of the human BBSome, comprising the subunits BBS1, 4, 5, 8, 9, and 18 (*Klink et al., 2017*). Although this complex lacks BBS2 and BBS7, it binds with up to sub-micromolar binding affinity to cargo proteins. In addition, we found that strongly binding ciliary trafficking motifs contain stretches of aromatic and positively charged residues, many of which were located in the third intracellular loop and the C-terminal domain of ciliary GPCRs. Multiple binding epitopes on cargo proteins cooperatively interact with multiple subunits of the BBSome (*Klink et al., 2017*), which is consistent with previous reports (*Jin et al., 2010*). A detailed insight of how these motifs interact with the BBSome requires a full molecular model of the complex to evaluate the potential interaction surfaces on the side-chain level.

Here we report the structure of the human BBSome core complex at an average resolution of 3.8 Å for BBS1, 4, 8, 9, and 18 and 4.3 Å for BBS5. The high quality of the map allowed us to build an atomic model of ∼80% of the complex. The structure reveals the architecture of the complex and the sophisticated intertwined arrangement of its subunits. A large positively charged region on the surface of the complex suggests how it orients on the negatively charged ciliary membrane. We found that the complex adopts a conformation in which the Arl6 binding site would be accessible. The high-resolution structure allows us to accurately locate pathogenic patient mutations and to decipher how they perturb intra- and/or intermolecular interactions on the molecular level. We identified a negatively charged cleft in the center of the complex, which is positioned favorably for cargo interactions. Based on these findings, we propose a model for how membrane-bound ciliary cargo proteins like GPCRs can bind to the BBSome.

## Results and discussion

### Architecture of the BBSome core

To obtain suitable samples for high-resolution structural analysis, we chose to work on the BBSome core instead of the full complex, because the core, comprising BBS1, 4, 5, 8, 9, and 18 was considerably better soluble, more stable and homogeneous, but still bound with high affinity to Arl6 and cargo peptides (*Figure 1A*) (*Klink et al., 2017*). We determined the structure of the BBSome core complex by electron cryo microscopy (cryo-EM) at an average resolution of 3.8 Å. Besides BBS5 that was only sub-stoichiometrically bound (*Figure 1—figure supplement 1A*), all domains were well resolved with only some connecting loops and N-and C-terminal regions missing (*Figure 1A*, *Figure 1—figure supplements 1* and *2*). After three-dimensional sorting (Materials and methods, *Figure 1—figure supplement 3*) we also resolved BBS5 at a resolution of 4.3 Å (*Figure 1—figure supplement 1*) and combined its density with the reconstruction of the full data set (*Figure 1B*) to build an atomic model (Materials and methods, *Figure 1C*, *Table 1*).

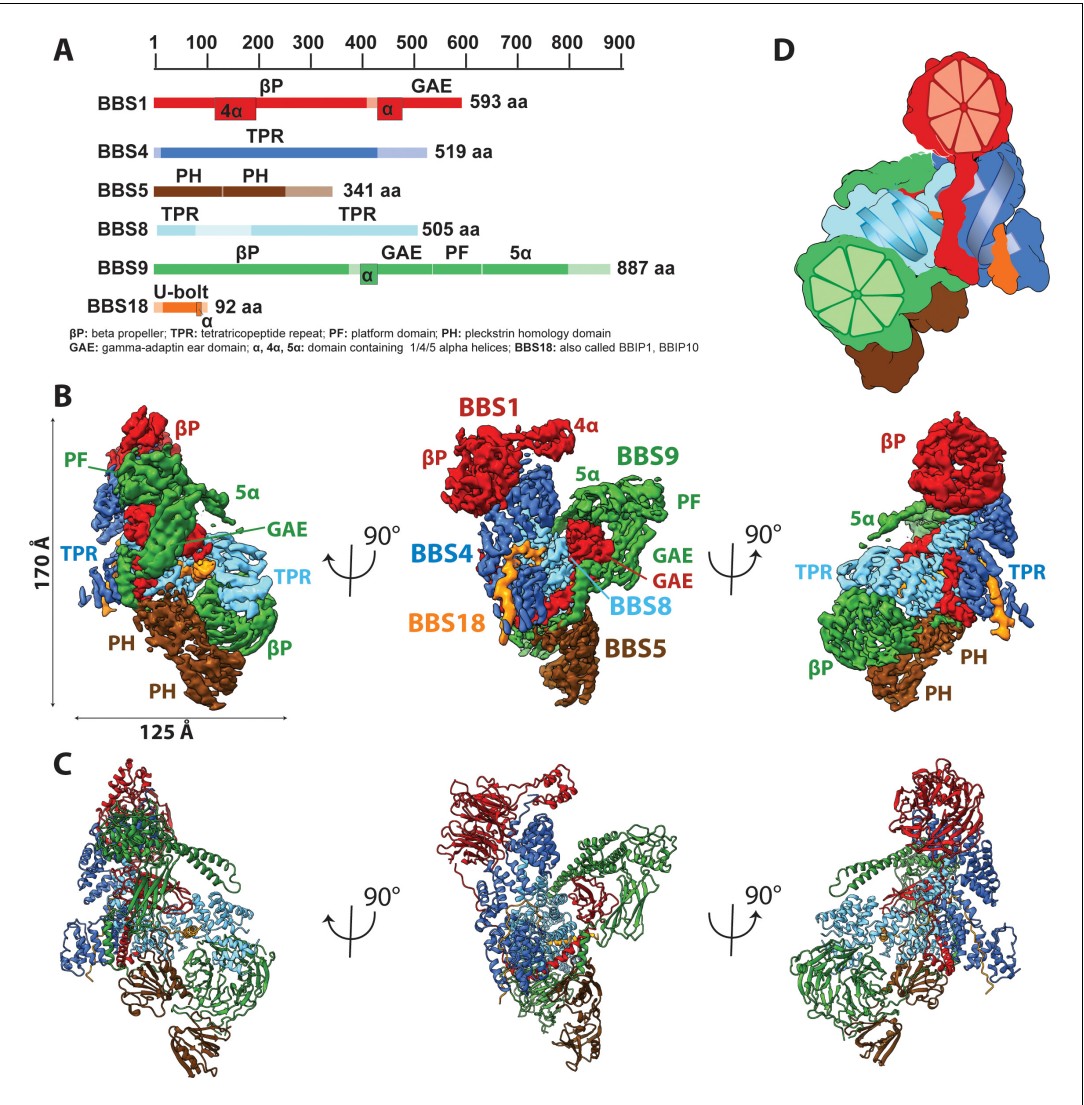

**Figure 1.** Architecture of the BBSome. (**A**) Domain architecture of the protomers forming the BBSome core complex. The parts of the primary structure that could be assigned in the density is shown in full colors, while non-modeled regions are represented in opaque colors. (**B**) Composite cryo-EM density map of the BBSome core complex in different orientations. Each protomer is colored differently and the thresholds of the segmented densities of the individual domains were adjusted to visualize each domain at an optimal signal intensity. BBS5 is poorly visible at the signal level of the other subunits and required a reconstruction from a subset of particles, which was obtained by 3D-sorting (*Figure 1—figure supplement 1*). (**C**) The final model of the core BBSome. (**D**): Schematic representation of the complex, highlighting the two β-propeller domains of BBS1 and BBS9 as red and green segmented wheels, and the super helical arrangements of the TPR repeats of BBS4 and BBS8 as blue and cyan helices.

The online version of this article includes the following figure supplement(s) for figure 1:

**Figure supplement 1.** Positioning of BBS5 in the BBSome core complex.
**Figure supplement 2.** Quality of the model of the BBSome core complex.
**Figure supplement 3.** Single particle processing workflow for structure determination of the human core BBSome.
**Figure supplement 4.** Sequence assignment of BBS18.
**Figure supplement 5.** Partially structured insert in BBS8, and hypothetical model for BBS9 dimerization.
**Figure supplement 6.** Surface potential and surface hydrophobicity of the core BBSome.

**Table 1.** EM data collection and refinement statistics of the core BBSome.

The BBS subunits 1,4,8,9 and 18 were modeled into the reconstruction from all particles that remained after ISAC 2D sorting (see blue box in *Figure 1—figure supplement 3*), while the subunit BBS5 was modeled into a map derived by 3D clustering from 180.654 out of 862.114 particles (see green box in *Figure 1—figure supplement 3*).

| Data collection | |
|---|---|
| Microscope | Titan Krios (Volta Phase plate, XFEG) |
| Voltage (kV) | 300 |
| Camera | K2 summit (Gatan) |
| Pixel size (Å) | 1.07 |
| Number of frames | 50 |
| Total electron dose (e⁻/Å²) | 67 |
| Number of particles | 2,831,329 |
| Defocus range (µm) | −0.3 – −1.0 |
| Phase Shift (degree) | 30–120 |
| **Atomic model composition** | **BBS 1,4,8,9,18 [+BBS5]** |
| Non-hydrogen atoms | 16,924 [+1175] |
| Protein atoms | 16,924 [+1175] |
| particle substack | 862,114 [180,654] |
| Ligand atoms | - |
| **Refinement (Phenix)** | |
| RMSD bond | 0.005 |
| RMSD angle | 0.993 |
| Model to map fit, CC mask | 0.79 |
| Resolution (FSC@0.143, Å) | 3.8 [4.3] |
| B-factor (Å²) | 145.45 |
| **Validation** | |
| Clashscore | 6.34 |
| Ramachandran outliers (%) | 0.09 |
| Ramachandran favoured (%) | 90.45 |
| Molprobity score | 1.89 |
| EMRinger score | 1.53 |

The BBSome core complex is arranged in multiple layers with its smallest 93 residue subunit BBS18 (also known as BBIP10, BBIP1) in its center (*Figure 2*). While being almost completely unfolded itself, BBS18 winds through two super-helically arranged TPR domains of the perpendicularly arranged BBS4 and BBS8 subunits, and clamps them together like a U-bolt (*Figure 1*, *Figure 2A,B*). The resulting Y-shaped arrangement forms the spine of the core BBSome complex. BBS1 binds to the N-terminal end of the TPR superhelix formed by BBS4 via its N-terminal β-propeller domain (*Figure 3A*), wraps around BBS4 and BBS8, and binds with its C-terminal GAE domain to the TPR domain of BBS8 (*Figure 2C*, *Figure 4A,C*). BBS9 has a similar domain architecture as BBS1, with two additional domains at its C-terminus. The N-terminal β-propeller of BBS9 binds to the N-terminus of the TPR superhelix formed by BBS8, in analogy to the BBS1/BBS4 interaction (*Figure 3B*). BBS9 wraps around BBS4 and parts of BBS1, and engulfs the GAE domain of BBS1 with its C-terminal GAE, platform and α-helical domains (*Figure 2D*, *Figure 4A*). BBS5 is composed of two PH domains that both interact with the β-propeller of BBS9. One of the PH domains also interacts with BBS8 and with an unstructured loop of BBS9 (*Figure 2E*; *Figure 1—figure supplement 1C,E*).

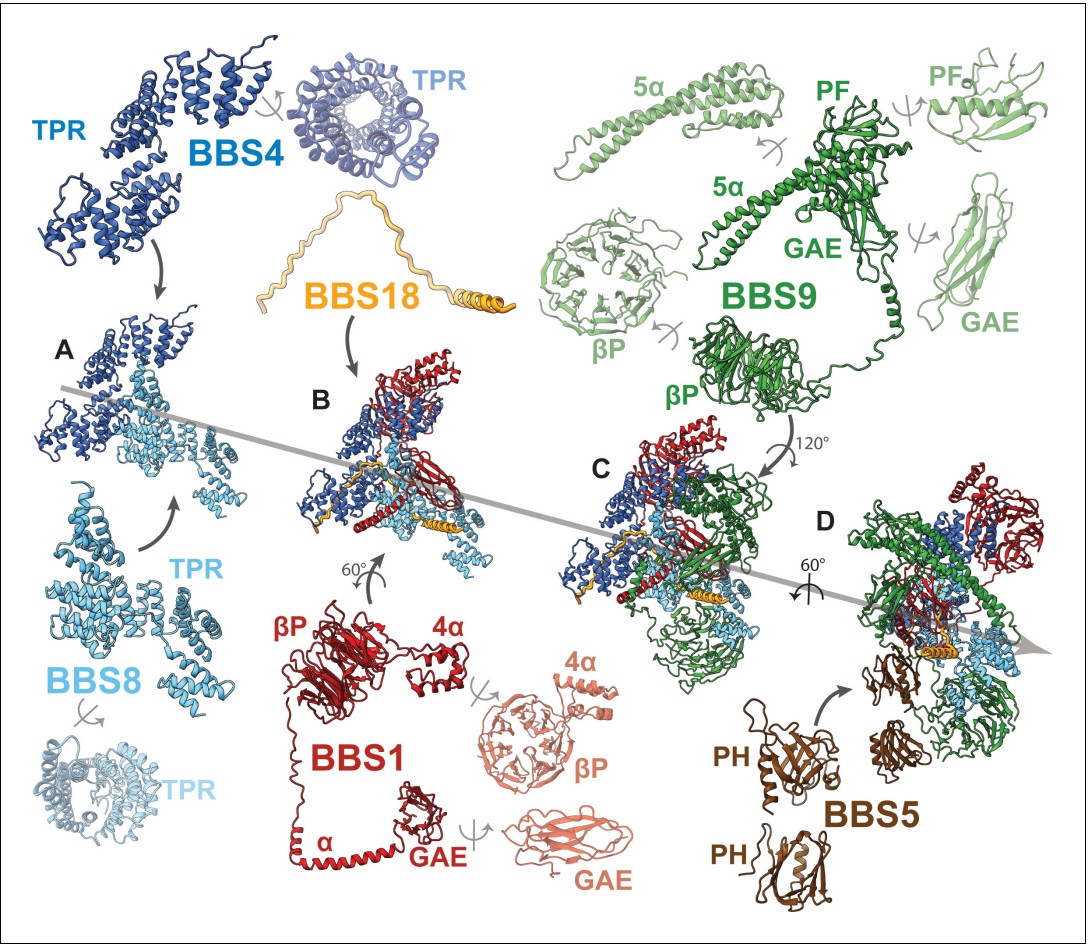

**Figure 2.** Arrangement of subunits and domains within the BBSome core. (**A–E**) The described order of addition of subunits has been chosen for visual clarity and does not reflect the sequential assembly in vivo. BBS4 and BBS8 are shown in two different views to visualize the superhelical arrangement of the TPR repeats. Likewise, domains of BBS1 and BBS9 are also shown individually in two views.

## BBS18 is a central stabilizing component of the core BBSome

From co-expression and pulldown studies of the individual BBSome subunits, we know that BBS18 is important for the stability of larger BBSome subcomplexes (*Klink et al., 2017*). The cryo-EM structure of the core BBSome delivers an explanation for these findings, revealing that BBS18 is involved in a large amount of stabilizing interactions with the TPR domains of BBS4 and BBS8 (*Figure 5A*, *Figure 6D–E*, *Figure 1—figure supplement 4I*). Because of its unfolded U-bolt region, BBS18 forms a large interaction surface with these subunits, resulting in high solvation free energies ($\Delta^i G$) (as analyzed by the Pisa server *Krissinel and Henrick, 2007*) (*Table 2*, *Figure 5A*, *Figure 1—figure supplement 4I*).

The short helix of BBS18 (residues L61 to N80) protrudes from the BBSome core complex and only loosely interacts with the GAE domain of BBS1 (*Figure 4D*) and with a loop of BBS8 (*Figure 1—figure supplement 5B*). Despite this limited interaction with other subunits, the helix is important for the proper assembly of the BBSome as has been shown previously, analyzing a disease-causing null mutation in BBS18 (L58*) which results in the loss of the helix (*Scheidecker et al., 2014*).

Taken together, these findings suggest that BBS18 functions as a structural scaffold protein that is essential for the proper assembly and structural stability of the BBSome complex.

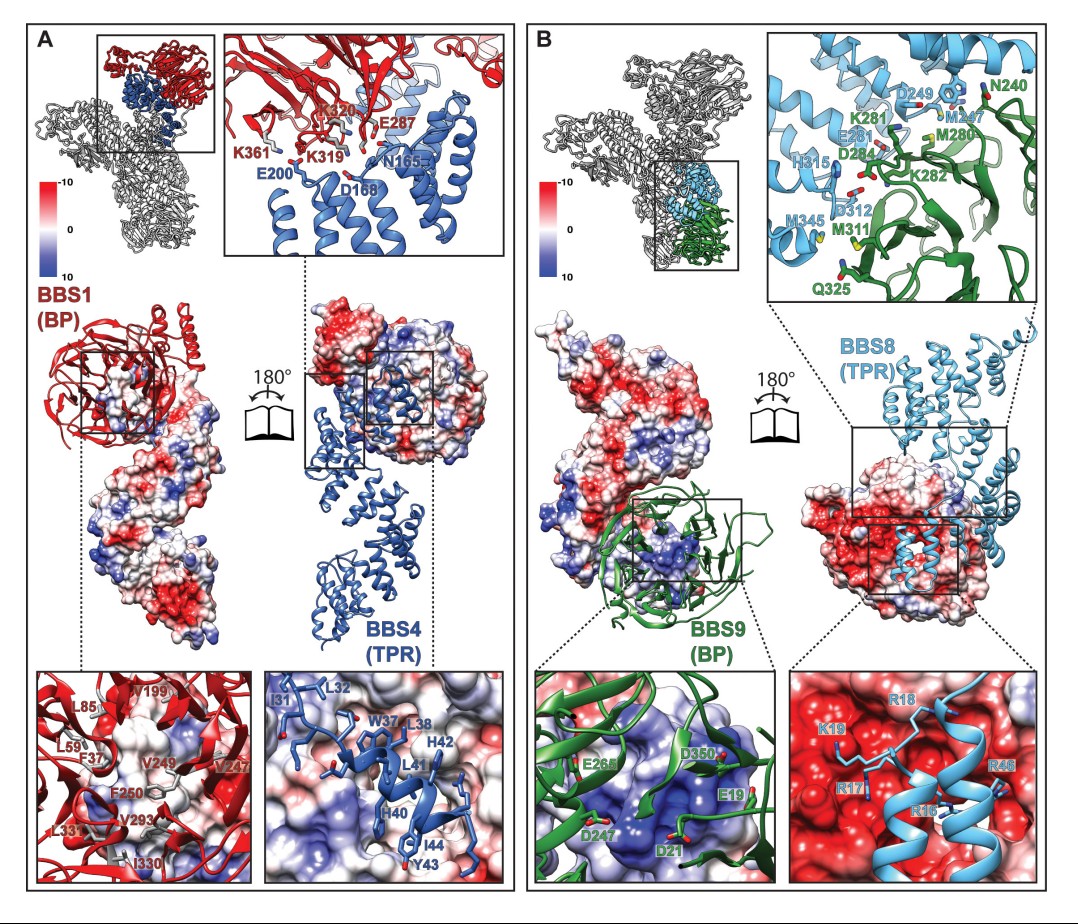

**Figure 3.** Interactions of the TPR repeat proteins BBS4 and BBS8 with β-propellers of BBS1 and BBS9, respectively. (**A**) Open book representation of the interaction surface of BBS4 with the β-propeller (BP) of BBS1. The interacting subunits are shown in ribbon and surface representations, respectively. Close-up views show relevant residues of the interaction surface. (**B**) Analogous open book representation of the interaction surface of BBS8 with the β-propeller (BP) of BBS9.

## Surface complementary β-propeller-TPR interactions

Interestingly, both structurally analogous TPR containing subunits BBS4 and BBS8 interact with β-propellers of BBS1 and BBS9, respectively. In both cases, it is the N-terminus of the TPR containing subunits that binds to the central grooves of the β-propellers (*Figure 3A,B*), the mode of interaction, however, differs.

As previously described (*Knockenhauer and Schwartz, 2015*), the β-propeller of BBS9 is negatively charged around its central cavity. Our structure reveals that it interacts with a positively charged patch of BBS8 (*Figure 3B*). The surface charges at the center of the β-propeller of BBS1 and the N-terminus of BBS4, however, are more evenly distributed and the interface at this position is mostly stabilized by hydrophobic interactions (*Figure 3A*). In addition, the superhelically arranged TPRs of BBS4 and BBS8 wind around and extensively interact with the side of the respective β-propeller. These interfaces are mostly stabilized by potential ionic interactions (*Figure 3A,B*). Moreover, the interface between BBS8 and the side of the BBS9 β-propeller contains two pairs of methionines, which potentially interact via hydrophobic and S/π interactions (*Figure 3B*).

Mutations at the BBS4-BBS1 and BBS8-BBS9 interfaces (*Table 3*), such as Q325R (*Chou et al., 2019*) at the side of the BBS9 β-propeller (*Figure 5B*) or E224K (*Redin et al., 2012*) and R268P (*Estrada-Cuzcano et al., 2012*) in the central cavity of the BBS1 β-propeller (*Figure 5C*) lead to disease in patients, indicating that proper interaction between these subunits is crucial for a functional BBSome.

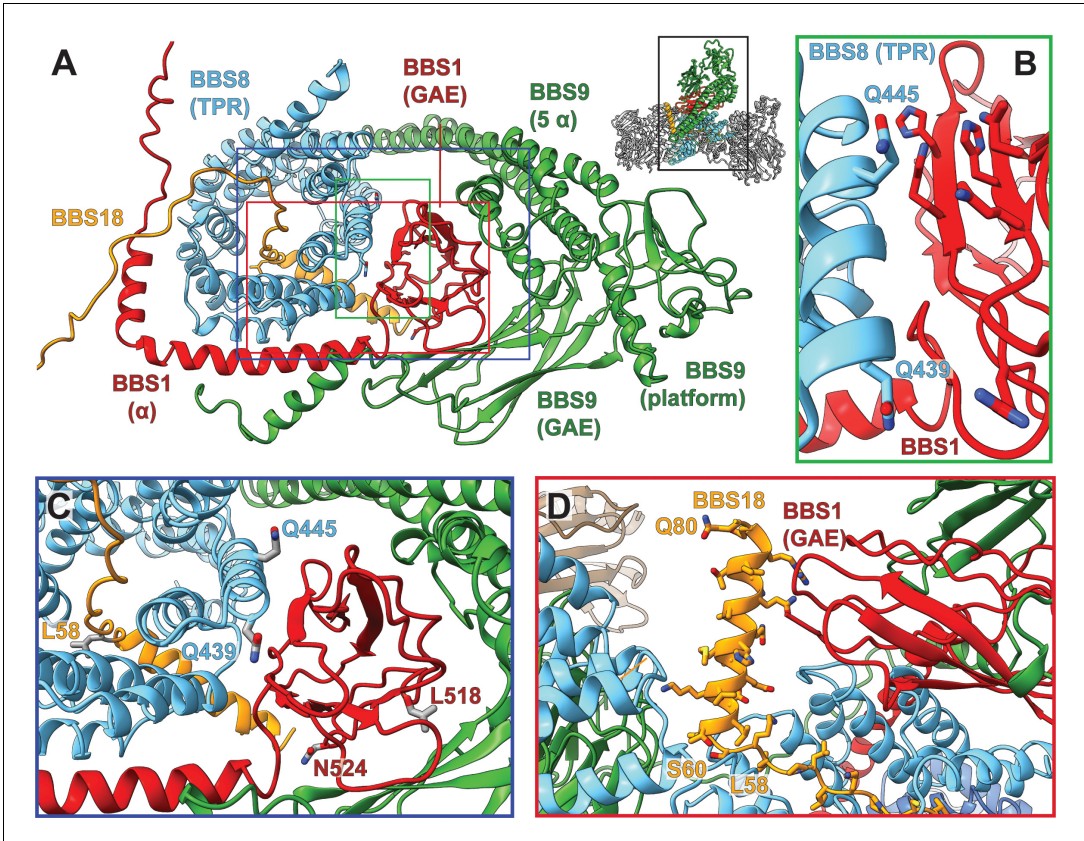

**Figure 4.** Pathogenic patient mutations within the interaction surface of BBS8 with BBS18 and with the C-termini of BBS1 and BBS9. (**A**) The GAE domain of BBS1 binds to BBS8 and is embraced by the C-terminus of BBS9. (**B,C**) Pathogenic patient mutations Q439H and Q445K on BBS8 and L518P and N524Δ on BBS1 disturb this interface. (**D**) In contrast, the C-terminal BBS18 helix (Ser60-Gln80) only interacts weakly with the complex, and is held in place by interactions with the GAE domain of BBS1. The pathogenic patient mutation L58* eliminates this helix.

In contrast to BBS4, BBS8 contains a long loop between tyrosine 53 and lysine 180, connecting the first N-terminal TPR domain with the rest of the TPR superhelix. This loop is not fully resolved in our structure, but can be partly visualized at lower resolution (*Figure 1—figure supplement 5A,B*). Starting after the first TPR domain, it winds through the center of the complex and back to the following TPR domains. It interacts with BBS18 and with several TPRs of BBS8 and does not interrupt the TPR superhelix (*Figure 1—figure supplement 5B*). Interestingly, the region around this loop varies in different human BBS8 splice variants, expressed in different tissues (*Murphy et al., 2015*; *Riazuddin et al., 2010*). For example, Exon 2a of BBS8 is only expressed in retina, resulting in an additional insert of 10 residues between helices 2 and 3 of BBS8, which is in direct proximity to the unmodeled density in our structure which we assign to the unstructured loop in BBS8 (*Figure 1—figure supplement 5B*). It is therefore conceivable that the loop in BBS8 is also involved in tissue-specific functions of the BBSome.

## The C-terminus of BBS9 forms a GAE binding motif

A prominent feature in the core BBSome structure is the expanded domain arrangement of BBS1 and BBS9 which allows the wrapping of these subunits around the central BBS4 and BBS8 subunits (*Figure 2C,D*). BBS1 and BBS9 interact with each other via their C-terminal domains (GAE, platform, and α-helical domain) in such a way that the β-propellers orient to the opposite direction locating to the periphery of the complex where they interact with the TPR domains of BBS4 and BBS8 (*Figure 2C,D*, *Figure 4A*). The interaction of BBS1 with BBS9 is essential for the function of the BBSome, as BBS1 mutations L518P (*Mykytyn et al., 2003*) and N524Δ (*Deveault et al., 2011*) at the interface between the GAE domain of BBS1 and BBS9 are pathogenic (*Figure 4*-C, *Table 3*). The

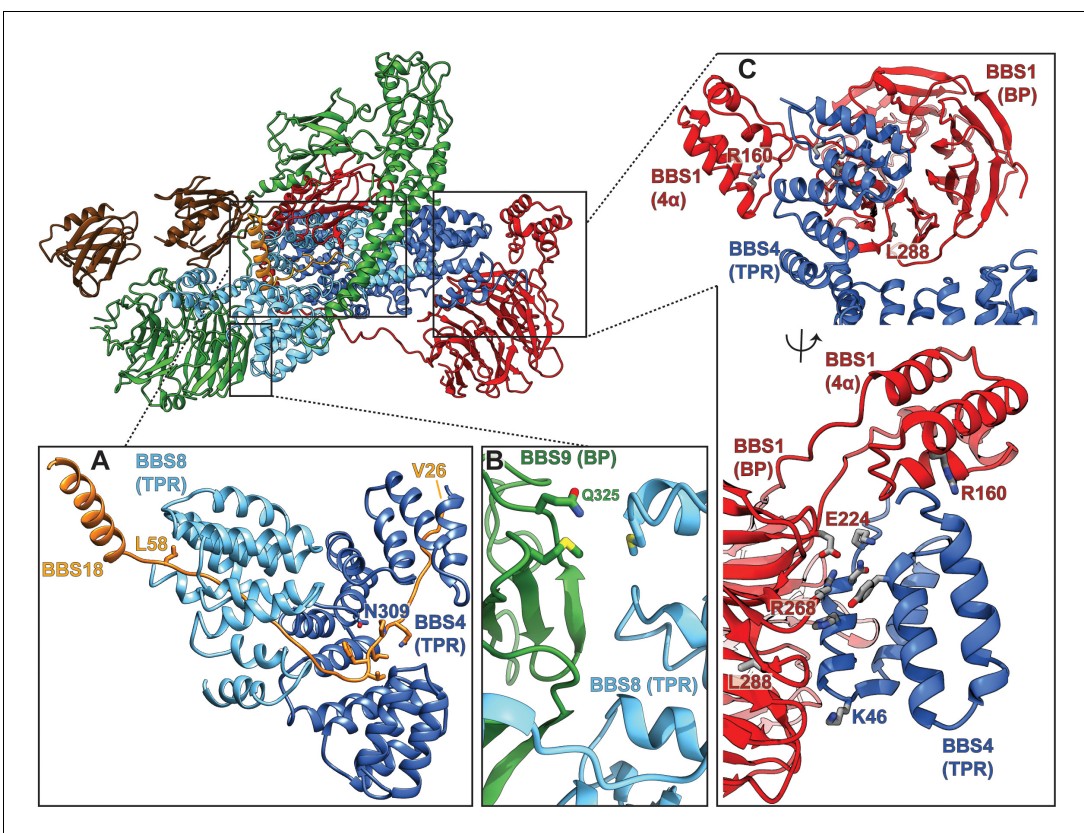

**Figure 5.** Pathogenic patient mutations within the core BBSome complex with the potential to disturb subunit interactions. (**A**) Interactions of BBS4 and BBS8 with the BBS18 'U-bolt' region (Val26-Lys59) stabilize the central spine of the complex. The mutation N309K disturbs a tight interaction of BBS4 with the main-chain of BBS18 in the U-bolt. The pathogenic patient mutation L58* eliminates the C-terminal helix of BBS18 but leaves the 'U-bolt' region that clamps together BBS4 and BBS8 intact. (**B**) The pathogenic mutation Q325R in BBS9 is located at the interface of the BBS9 β-propeller with BBS8. (**C**) Several pathogenic patient mutations are located at the interface of the BBS1 β-propeller and its helical insertion with the N-terminus of BBS4 (R160Q, E224K, R268P, L288R), underlining the importance of this interface.

GAE domain of BBS1 is also involved in interactions with BBS8. The glutamine residues 439 and 445 in BBS8 are located at this interface and their mutation results in disease (Q439H, Q445K) (*Chou et al., 2019*; *van Huet et al., 2015*; *Ullah et al., 2017*; *Goyal et al., 2016*) (*Figure 4C*, *Table 3*).

The GAE domains of BBS1 and BBS9 interact strongly with each other (*Figure 4A*). Since their structure is very similar, it can be imagined that these domains would induce self-dimerization of either BBS1 or BBS9. Indeed, in previous studies (*Knockenhauer and Schwartz, 2015*; *Klink et al., 2017*), it was shown that the isolated C-terminal region of BBS9, including the GAE domain, forms a dimer in solution. A superposition of BBS9 monomers reveals that an interaction via their GAE domains would not result in steric clashes (*Figure 1—figure supplement 5C*). Taken together with our observation that the GAE domains strongly interact in the BBSome core complex, we propose that the strong heterodimeric interaction between the GAE domains is the reason for isolated BBS9 to homodimerize via its GAE domain.

## BBSome subcomplexes bind to phosphoinositides in the absence of BBS5

Compared to the other subunits, BBS5 is more loosely attached to the BBSome core complex, as we could only identify the subunit in a subpopulation of particles (*Figure 1—figure supplements 1* and *3*). Similarly, BBS5 was also missing from a subset of natively purified BBSome complexes

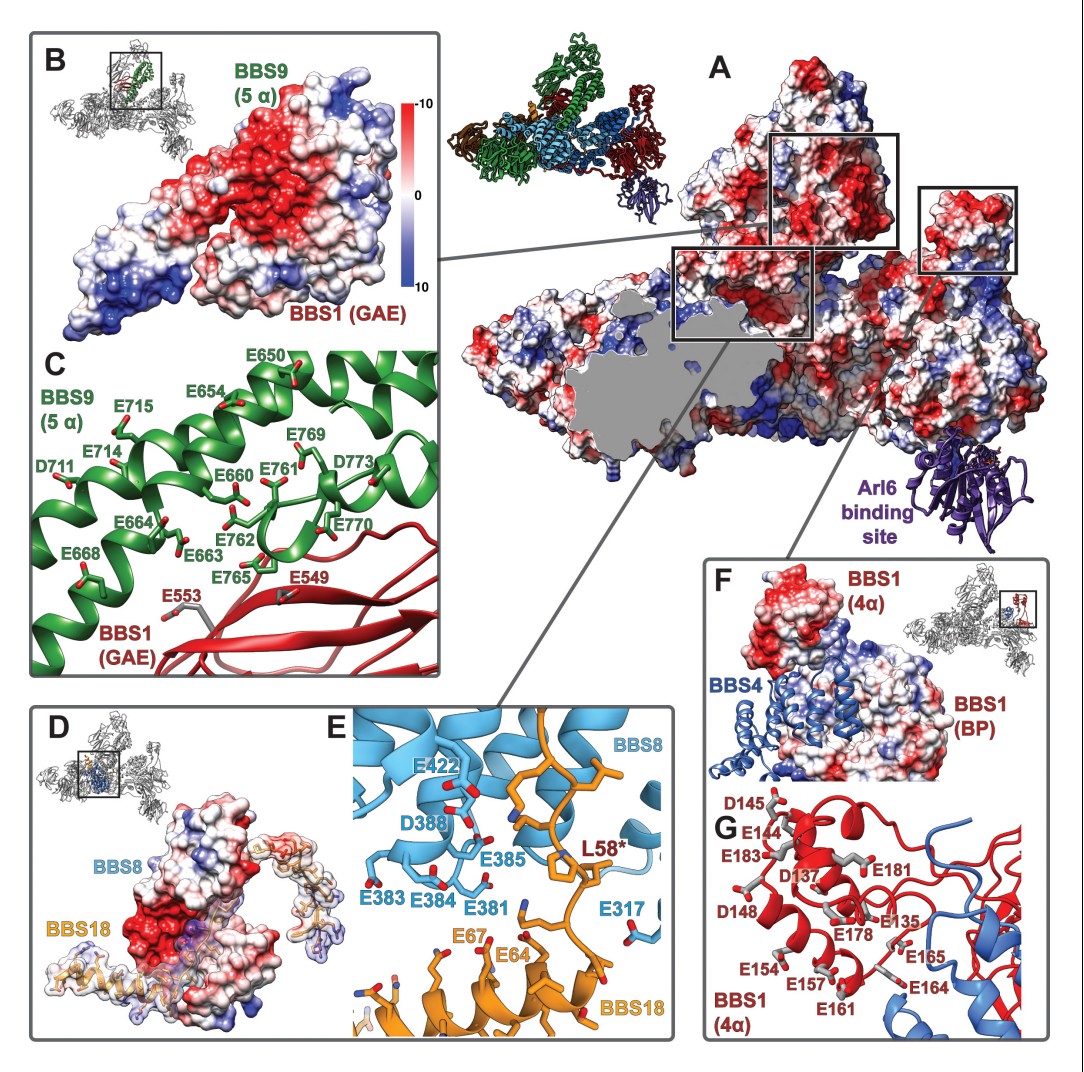

**Figure 6.** Highly negatively charged regions in the central cleft of the BBSome. (**A**): An open cleft within the center of the core BBSome contains multiple highly negatively charged regions that might be involved in cargo binding. The Arl6 binding site, as deduced from the crystal structure of the β-propeller of BBS1/Arl6 (*Mourão et al., 2014*), is shown as purple ribbon. (**B,D,F**): Surface charges in three hotspots of negative charge within the central cleft. (**C, E,G**): Positions of negatively charged residues.

(*Chou et al., 2019*). In addition, the resolution is limited in this region, indicating a high degree of conformational flexibility.

BBS5 is composed of two pleckstrin homology (PH) domains that were predicted to be close structural relatives to the PH-like domains PH-GRAM (*Begley et al., 2003*) and GLUE (*Teo et al., 2006*). In the BBSome core structure, the PH domains are laterally rotated by 90° to each other (*Figure 2E*). Like PH-GRAM and GLUE, BBS5 was shown to bind phosphoinositides, with the highest preference for phosphatidylinositol 3-phosphate (PI(3)P) and phosphatidic acid (PA), which was suggested to be crucial for ciliogenesis (*Nachury et al., 2007*). Besides BBS1 that indirectly interacts with membranes via Arl6 (*Jin et al., 2010*; *Mourão et al., 2014*), BBS5 likely mediates the contact to membranes by direct interactions with phosphoinositides (*Nachury et al., 2007*). The specific binding to certain PIPs is a potential way to regulate BBSome transportation in and out of the cilium, as the composition of ciliary membranes differs from that of the plasma membrane (*Garcia-Gonzalo et al., 2015*; *Chávez et al., 2015*; *Nachury and Mick, 2019*).

To find out whether BBS5 is the only subunit of the BBSome complex that interacts with phosphoinositides, we compared the binding of different phosphoinositides to the core BBSome

**Table 2.** Interfaces between core BBSome subunits.
The mutual subunit interaction surfaces within the core BBSome with highest relevance for complex stability (i.e. with solvation free energies ($\Delta^i$G)$<-2.5$ kcal/mol and with interface areas $> 400$ Å$^2$), as analyzed by the Pisa server (*Krissinel and Henrick, 2007*).

| BBSome subunit A | BBSome subunit B | Interface area (A-B) [Å$^2$] | $\Delta^i$G (A-B) [kcal/mol] |
|---|---|---|---|
| BBS18 | BBS8 | 1621.0 | −28.1 |
| BBS18 | BBS4 | 1411.0 | −25.4 |
| BBS9 | BBS1 | 1946.2 | −23.0 |
| BBS9 | BBS8 | 2148.3 | −22.7 |
| BBS8 | BBS1 | 1785.8 | −20.5 |
| BBS4 | BBS1 | 2429.1 | −18.0 |
| BBS8 | BBS4 | 1469.4 | −13.1 |

complex containing BBS 1, 4, 5, 8, 9 and 18 with a smaller complex that lacks BBS5 and BBS1 (*Figure 7A–C*). We found that the core BBSome complex interacts preferably with PI(3)P, PI(3,5)P$_2$, PI(4,5)P$_2$, PI(5)P, and PA. This pattern is similar to the one observed for BBS5 alone (*Nachury et al., 2007*), however, the interaction with PI(3,5)P$_2$ and PI(4,5)P$_2$ is more pronounced in the case of the BBScome core complex (*Figure 7B*). Surprisingly, the smaller 4mer complex also binds specifically to the same phosphoinositides as the BBS5 and BBS1-containing complex and especially strong to PA (*Figure 7A*). This shows that BBS5 is not exclusively responsible for phosphoinositide and PA binding and that respective binding sites must exist on one or more of the other subunits, namely BBS 4, 8, 9 or 18.

## Functional subcomplexes of the BBSome

There are indications that subcomplexes of the BBSome exist and that these might have functional relevance (*Barbelanne et al., 2015*). The BBS5 subunit was shown to be dispensable for BBSome assembly (*Zhang et al., 2012a*), and was only found in a subset of particles both from our core BBSome purification and in a native purification of the full bovine BBSome complex (*Chou et al., 2019*). Although this could also be explained by a misfolded or denatured BBSome, such substoichiometric complexes would be consistent with the existence of functional complexes lacking BBS5.

**Table 3.** Disease-causing variants at the interface between BBSome subunits.
Only mutations that sit at the interface between BBSome subunits and likely have an influence on the stability of the complex have been analyzed.

| BBS gene | mutation | phenotype | Reference |
|---|---|---|---|
| BBS1 | L518P | BBS | *Mykytyn et al., 2003* |
| | N524Δ | BBS | *Deveault et al., 2011* |
| | R160Q | BBS, RP | *Sharon and Banin, 2015*; *Deveault et al., 2011* |
| | E224K | BBS | *Redin et al., 2012* |
| | R268P | BBS | *Estrada-Cuzcano et al., 2012* |
| | L288R | BBS | *Muller et al., 2010* |
| BBS4 | N309K | BBS | *Muller et al., 2010* |
| BBS8 | Q439H | BBS, RP | *Ullah et al., 2017*; *Goyal et al., 2016* |
| | Q445K | RP | *van Huet et al., 2015* |
| BBS9 | Q325R | BBS | *Chou et al., 2019* |
| BBS18 | L58* | BBS | *Scheidecker et al., 2014* |

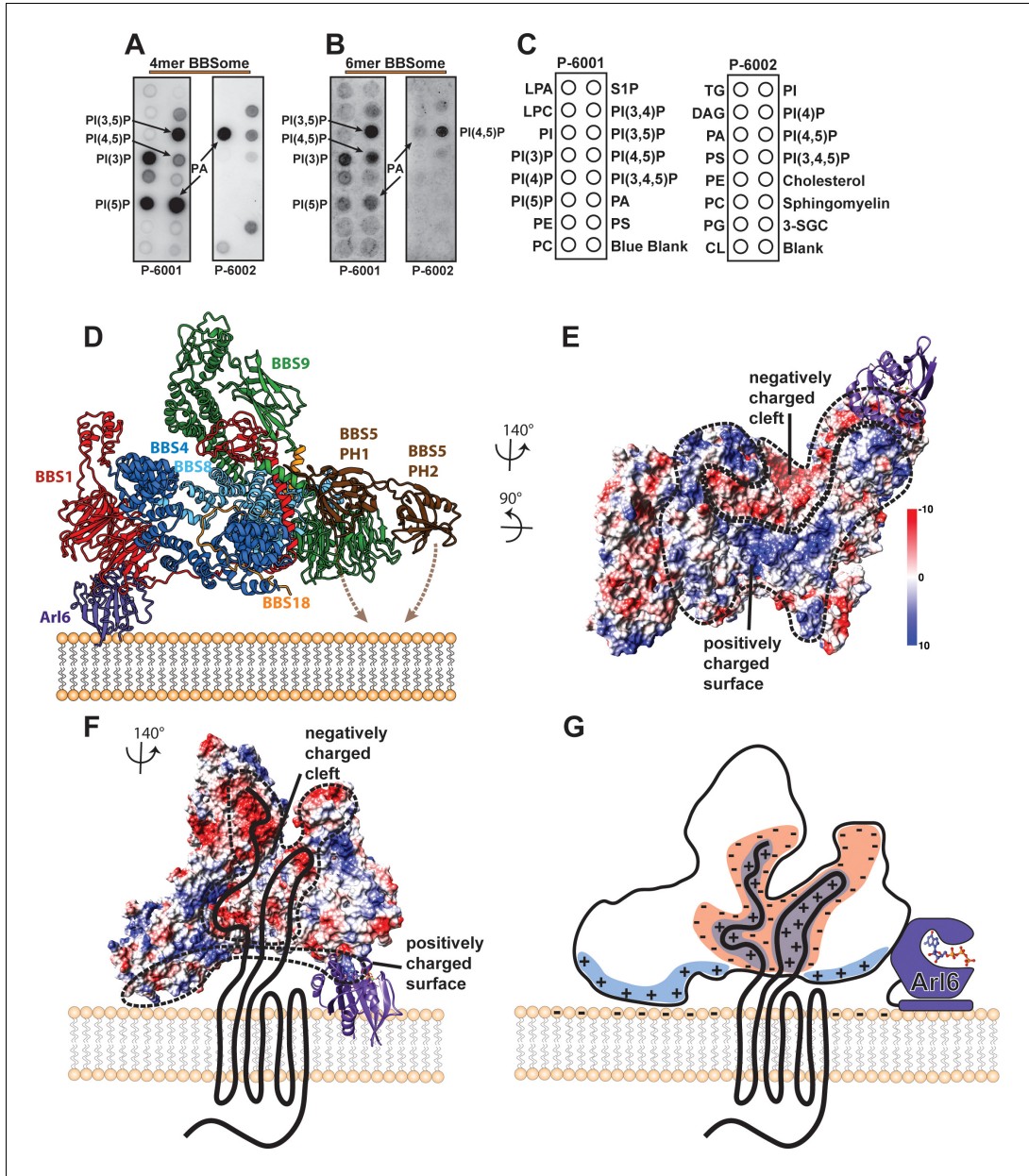

**Figure 7.** Interaction of the core BBSome with membranes. (A–C) The affinity of the 4mer BBSome complex containing BBS4, 8, 9 and 18 (A) and of the core BBSome complex containing BBS1, 4, 5, 8, 9, 18 (B) to different lipids were probed in a protein-lipid overlay assay. For this, hydrophobic membranes with immobilized lipids as depicted in (C) (so-called 'PIP-strips', Echelon) were blocked with TBS-T + 3% fatty acid–free BSA and then incubated with 7.5 μg/ml complex for one hour at room temperature. After washing three times with TBS-T + 3% fatty acid–free BSA, immobilized complexes were detected by Western blot against the Flag-tag on BBS8. The PIP strip experiments indicate that BBSome subcomplexes interact specifically with PIPs even in the absence of both the Arl6-binding subunit BBS1 and the previously described PIP-binding subunit BBS5 (40). (D): Potential orientation of the BBSome core complex towards the membrane. The orientation of Arl6 towards the BBSome was deduced from the crystal structure of the β-propeller of BBS1/Arl6 (*Mourão et al., 2014*), which was overlaid with the BBS1 β-propeller from the BBSome core complex. In such an arrangement, a positively charged surface of the core complex would be oriented towards the membrane (E,G) and a negatively charged cleft in the vicinity of BBS1 is favorably positioned to accept BBSome-binding regions from cargo proteins like GPCRs (E–G), which were found to be mostly positively charged (*Klink et al., 2017*). A model how GPCRs might be recognized by the BBSome is depicted in (G).

The online version of this article includes the following figure supplement(s) for figure 7:

*Figure 7 continued*
**Figure supplement 1.** Comparison of bovine BBSome with human core BBSome.

Likewise, both in native and in our heterologous expressions, the top lobe formed by BBS2 and BBS7 readily dissociates from the complex (*Chou et al., 2019*) or renders it insoluble (*Klink et al., 2017*). Chou et al. suggested that the stability of the α-hairpin in the C-terminal domain of BBS9 likely depends on the stabilization by its partner hairpin in BBS2. We found that in the absence of BBS2, the BBS9 hairpin folds towards BBS8 and is stabilized that way (*Figure 7—figure supplement 1D,E*). This further supports our hypothesis that the core BBSome is an independently stable entity (*Klink et al., 2017*), and that BBS2 and BBS7 (and also BBS5) are probably not required in every step within the 'lifecycle' of the BBSome.

## Membrane association and cargo recognition of the core BBSome

The BBSome is recruited to membranes by the small GTPase Arl6 in a GTP-dependent manner (*Jin et al., 2010*). A crystal structure of Arl6 with the β-propeller of BBS1 from *Chlamydomonas reinhardtii* revealed how Arl6 interacts with the BBSome in atomic detail (*Mourão et al., 2014*). To analyze how the core BBSome would orient on the membrane, we placed Arl6 at the same position as in the crystal structure by overlaying the respective BBS1 β-propeller domains (*Figure 7D*). The BBS1 β-propeller is located at the periphery of the core BBSome complex, and – in the absence of BBS2 and BBS7 - is freely accessible to bind to membrane-attached Arl6. Such an orientation would position a positively charged surface patch of the core BBSome close to the membrane, which is favorable for interactions with the negatively charged membrane surface. Importantly, this orientation also leaves a negatively charged cleft in the BBSome structure oriented perpendicular to the plane of the membrane (*Figure 7E,F*). We have previously found that prominent features of GPCR cargo molecules which determine binding to the core BBSome are motifs composed of aromatic and basic residues, many of which are found in the third intracellular loop and the C-terminal tail of GPCRs (*Klink et al., 2017*). The negatively charged cleft within the BBSome probably interacts specifically with these positively charged motifs and is thereby directly involved in cargo recognition (*Figure 7E–G*). This would also position the motifs close to BBS1, a subunit which was shown to be particularly important for cargo recognition (*Jin et al., 2010*; *Seo et al., 2011*; *Su et al., 2014*; *Seo et al., 2009*; *Bhogaraju et al., 2013*; *Ruat et al., 2012*; *Zhang et al., 2012b*).

The negative charge in the cleft is not formed by BBS1 alone, but also by residues of all other core BBSome subunits, except BBS5. There are three 'hotspots' of negative charge. One of them is located at the contact surface of the GAE domain of BBS1 and the 5α domain of BBS9 (*Figure 6B, C*), the second one is deeply buried in the core BBSome complex at the junction point where the BBS18 U-bolt ends and descends into a short α-helix (*Figure 6D,E*), and the third one is determined by the α-helical insertion within the BBS1 β-propeller, which contains many glutamate or aspartate residues (*Figure 6F,G*). The α-helical insertion can only get in contact with cargo peptides that extend far into the negatively charged cleft (*Figure 7D*). The β-propeller itself and the C-terminal GAE domain of BBS1 are also accessible from within the cleft. They have a more balanced charge distribution and might contribute to cargo binding by hydrophobic and ionic interactions at the base of the cleft, which would explain the high importance of BBS1 for cargo interactions.

The large size of the binding cleft with different 'hotspots' suggests that cargo recognition is likely very complex and variable, involving the interaction with different BBSome subunits. The proper study of BBSome-cargo interactions therefore requires the full BBSome complex as cargo binding to single subunits, albeit relevant, does not take interaction to multiple sites into account. For example, we previously identified a peptide fragment from the C-terminal part of SSTR3 which binds to the core BBSome with ~100 fold higher affinity than to the isolated BBS1 β-propeller (aa1-430) (*Klink et al., 2017*).

Since the BBS5 and Arl6 binding sites are located on opposite sides of the core BBSome complex (*Figure 7D*), a simultaneous membrane binding of both motifs would require a curved membrane surface, as it is present on the inner cilium wall, and/or conformational rearrangements of the complex. The observed dynamic properties of BBS5 (*Figure 1—figure supplements 1* and *3*) are

probably the prerequisite for the subunit to be able to reorient and position one or both of its PH domains closer to the membrane, thereby enabling interaction with the membrane.

While this manuscript was being prepared, the structure of the bovine BBSome complex at 4.9 Å was published, in which the Arl6 binding site is blocked by the subunits BBS2 and BBS7 (25). These two additional subunits form a highly intertwined lobe that contacts the BBSome core at the C-terminal hairpin in BBS9 and at the β-propeller of BBS1. While this blocks the binding site for Arl6, the complex was still able to bind to immobilized Arl6 during the purification, indicating a high degree of conformational dynamics of the lobe formed by BBS2 and BBS7 (25). The overall architecture of both BBSome complexes is very similar, but interestingly, in a rigid body overlay of our structure with the bovine BBSome, the Arl6 binding site does not clash with BBS2 or BBS7 (*Figure 7—figure supplement 1*). This is due to a different orientation of the BBS1 β-propeller to which Arl6 binds. In the structure of the bovine BBSome complex the BBS7 β-propeller interacts with the β-propeller of BBS1 and thereby pulls it towards the BBS2-BBS7 lobe. The small change of the BBS1 β-propeller orientation results in a ~ 20° change in the angle of Arl6 binding to the complex (*Figure 7—figure supplement 1F–H*).

While we cannot rule out that the changed BBS1 β-propeller adopts a non-physiological conformation in the absence of BBS2 and BBS7, it is probable that the reorientation of the propeller will occur in a similar manner when the highly dynamic BBSome 'top lobe' formed by BBS2 and BBS7 opens up to allow Arl6 binding (*Chou et al., 2019*). We therefore believe that our BBSome core structure represents the open conformation of the complex in contrast to the autoinhibited closed state of the bovine BBSome complex (*Chou et al., 2019*).

The additional subunits BBS2 and BBS7 extend the interaction surface that we suggest to bind to ciliary membranes (*Figure 7E*, *Figure 7—figure supplement 1C*), and narrow down the opening to the negatively charged cleft. However, it is still sufficiently large for peptides from cargo proteins to enter the cleft (*Figure 7F,G*, *Figure 7—figure supplement 1C*).

### Future perspectives

Future studies should reveal the details of how cargo proteins get recognized and how the 'top lobe' formed by BBS2 and BBS7 makes space for cargo binding, particularly in the context of BBSome interactions with Arl6 and the IFT complex. For this it will be crucial to obtain structures of the BBSome complex bound to different cargo peptides or full cargo proteins. The architecture of the extended negatively charged cleft within our structure of the core BBSome suggests that cargo binding depends on the 3-dimensional arrangement of all BBSome subunits, and that different cargo proteins might utilize different binding modes within the cleft. This would also suggest that the BBSome complex binds differently to cargo proteins in its open or its closed conformation.

Other intriguing questions to be addressed in the future include determining the precise orientation of the BBSome on membranes, and the relevance of interactions with phosphoinositides via BBS5 or via currently unidentified phosphoinositide binding motifs on one of the subunits BBS4, 8, 9 or 18 (compare *Figure 7A–C*).

## Materials and methods

**Key resources table**

| Reagent type (species) or resource | Designation | Source or reference | Identifiers | Additional information |
|---|---|---|---|---|
| Gene (*Homo sapiens*) | BBS1 | N/A | NCBI reference sequence: NM_024649.4 | |
| Gene (*Homo sapiens*) | BBS4 | N/A | NCBI reference sequence: NM_033028.3 | |
| Gene (*Homo sapiens*) | BBS5 | N/A | NCBI reference sequence: NM_152384.2 | |
| Gene (*Homo sapiens*) | BBS8 | N/A | NCBI reference sequence: NM_198309.2 | |

*Continued on next page*

*Continued*

| Reagent type (species) or resource | Designation | Source or reference | Identifiers | Additional information |
|---|---|---|---|---|
| Gene (*Homo sapiens*) | BBS9 | N/A | NCBI reference sequence: NM_198428.2 | |
| Gene (*Homo sapiens*) | BBS18 | N/A | NCBI reference sequence: NM_001195306.1 | |
| Cell line (*S. frugiperda*) | SF9 | Thermo Fisher (Germany) | RRID:CVCL_0549 | |
| Cell line (*Trichoplusia ni*) | Hi5 | Thermo Fisher (Germany) | RRID:CVCL_C190 | |
| Antibody | Mouse monoclonal anti-Flag antibody | Thermo Fisher (Germany) | Cat.No. MA1-91878 | dilution 1:5000 |
| Other | ACEMBL | *Vijayachandran et al., 2011* PMID: 21419851 | | Recombinant expression system for multiprotein complexes |
| Software, algorithm | SPHIRE software package | *Moriya et al., 2017* PMID: 28570515 | | |
| Software, algorithm | CrYOLO | *Wagner et al., 2019* PMID: 31925080 | | |
| Software, algorithm | Chimera | *Pettersen et al., 2004* PMID: 15264254 | | |
| Software, algorithm | Coot | *Emsley et al., 2010* PMID: 20383002 | | |
| Software, algorithm | Phenix | *Adams et al., 2010* PMID: 20124702 | | |
| Software, algorithm | RaptorX | *Wang et al., 2018* PMID: 28845538 | | |
| Software, algorithm | HHpred | *Söding et al., 2005* PMID: 15980461 | | |
| Software, algorithm | iMODFIT | *Lopéz-Blanco and Chacón, 2013* PMID: 23999189 | | |

## Purification of BBSome subcomplexes

The core BBSome (BBS 1, 4, 5, 8, 9, 18) and the 4mer BBSome subcomplex (BBS 4, 8, 9, 18) were cloned and purified as described previously (*Klink et al., 2017*). An SDS-PAGE analysis of a batch of both complexes, which was one of the batches that was also used for this study, can be assessed in *Figure 1A* of the manuscript (*Klink et al., 2017*). Briefly, the proteins were overexpressed in Hi5 insect cells (Thermofisher Scientific), harvested by centrifugation at 3000 rpm, and resuspended in lysis buffer (50 mM Hepes pH 8.0, 150 mM NaCl, 5 mM $MgCl_2$, 10% glycerol, 1 mM PMSF, 1 mM benzamidine, 1 mM TCEP). The resuspended cells were lysed with a Dounce Tissue Grinder (Wheaton, Millville, NJ), cell debris was removed by centrifugation at 25.000 rpm, and the supernatant was used for further protein purification.

For affinity purification, the cell supernatant was loaded on a column filled with Strep-Tactin Superflow high capacity resin (IBA). The column was washed and the proteins were eluted with elution buffer (50 mM Hepes pH 8.0, 150 mM NaCl, 5 mM $MgCl_2$, and 10% glycerol, 1 mM TCEP, and 10 mM D-desthiobiotin (IBA)). The eluted protein complexes were further purified by $Ni^{2+}$–NTA (Qiagen, Germany) affinity and/or anti-Flag M2 affinity chromatography (Sigma, Germany). Affinity-purified proteins were subjected to size-exclusion chromatography on a Superdex 200 10/300 column or on a Superose 6 5/150 column (GE Healthcare, Germany) in gel filtration buffer (50 mM Hepes pH 8.0, 150 mM NaCl, 5 mM $MgCl_2$, 10% glycerol and 0.1 mM TCEP).

## Protein-Lipid overlay assays

The affinity of the core BBSome complex (BBS1, 4, 5, 8, 9, 18) and the 4mer BBSome complex containing BBS4, 8, 9 and 18 to different lipids immobilized on hydrophobic membranes ('PIP-strips'

P-6001, P6002, Echelon) were probed in a protein-lipid overlay assay according to the manufacturer's instructions. In brief, PIP-strips were blocked with TBS-T + 3% fatty acid–free BSA and then incubated with 7.5 µg/ml complex for one hour at room temperature. After washing three times with TBS-T + 3% fatty acid–free BSA, immobilized complexes were detected by a western blot against the Flag-tag on BBS8.

## Chemical crosslinking to improve complex stability

To improve the stability of the core BBSome complex for cryo-EM experiments, the purified protein was diluted to 0.1 mg/ml in gel filtration buffer and treated with 0.05% glutaraldehyde at 20°C for two minutes. The reaction was stopped by adding TRIS buffer to a concentration of 100 mM. The protein was then concentrated and subjected to size-exclusion chromatography on a Superose 6 5/150 column (GE Healthcare, Germany) in plunging buffer (20 mM TRIS-HCl pH 7.5, 150 mM NaCl and 0.1 mM TCEP) to remove particles that aggregated during the crosslinking procedure. Fractions containing the crosslinked BBSome complex were immediately used for cryo-EM experiments.

## Sample vitrification

The concentration of the core BBSome as derived from chemical crosslinking was adjusted to 0.05–0.08 mg/ml and 4 µl of sample was immediately applied to a glow-discharged UltrAuFoil R1.2/1.3 holey gold grid (Quantifoil). After two minutes incubation at 13°C and 100% relative humidity (RH), the sample was manually blotted, replaced by fresh 4 µl sample, and then automatically blotted and plunged in liquid ethane using a Vitrobot (FEI).

## Electron microscopy and image processing

We observed the tendency of the BBSome complex to partially dissociate, which lead to a significant proportion of incomplete particles in many images, particularly in those with very thin ice, probably indicating some denaturation at the air-water interface. The problem persisted even after crosslinking and subsequent gel filtration. In addition, the particles had the tendency to aggregate, which further complicated the identification of intact single particles at the low contrast of typical EM micrographs, particularly at low defocus. We therefore pursued the use of a Volta phase plate (VPP) for data collection to enhance image contrast in order to improve the selection of complete complexes later on *Danev et al. (2014)*.

Cryo-EM datasets were collected on a Titan Krios electron microscope (FEI) equipped with a post-column energy filter, a Volta phase plate (VPP) and a field emission gun (FEG) operated at 300 kV acceleration voltage. A total of 15,266 micrographs were recorded on a K2 direct electron detector (Gatan) with a calibrated pixel size of 1.07 Å. The energy filter was used for zero-loss filtration with an energy width of 20 eV. In total 50 frames (each 300 ms) were recorded, resulting in a total exposure time of 15 s and a total electron dose of 67 e$^-$Å$^{-2}$. Data was collected using the automated data collection software EPU (FEI), with a defocus range of −0.3 to −1.0 µm. The position of the VPP was changed every 60 to 120 images, resulting in phase shifts of 30–120 degrees in >95% of all micrographs. Beam-induced motion was corrected for by using Motioncor2 (*Zheng et al., 2017*) to align and sum the 50 frames in each micrograph movie and to calculate dose-weighted and unweighted full-dose images. CTF parameters were estimated from the unweighted summed images and from micrograph movies utilizing the 'movie mode' option of CTFFIND4 (*Rohou and Grigorieff, 2015*). For subsequent steps of data processing using the software package SPHIRE/EMAN2 (*Moriya et al., 2017*), dose-weighted full dose images were used to extract dose-weighted and drift-corrected particles with a final window size of 280 × 280 pixels.

A total of 10 datasets were collected from several independently prepared protein samples and were successively processed by a combination of manual and automated particle extraction using crYOLO (*Wagner et al., 2019*), 2D sorting using the iterative stable alignment and clustering (ISAC) as implemented in SPHIRE, and merging with existing data to consecutively improve the size and quality of the derived particle stack. An initial model for 3D refinement was generated from the ISAC 2D class averages using RVIPER from SPHIRE, and was used as input for the first 3D refinement using MERIDIEN (3D refinement in SPHIRE). The obtained 3D reconstruction of MERIDIEN was then sharpened and filtered to its nominal resolution, and used as input for subsequent 3D refinements. Completeness of data collections was evaluated based on the improvements in resolution of the

resulting final reconstructions, which plateaued at 4 Å resolution from a cleaned set of 724,828 particles, as estimated by the 'gold standard' criterion of FSC = 0.143 between two independently refined half maps.

To further improve the quality of the derived particle stack, an alternative processing attempt was tested in which all 15266 micrographs were combined and subjected to automatic particle picking using crYOLO, selecting 1,973,261 particles with a high confidence score (threshold 0.7) and additional 858,068 particles that had a lower score (threshold greater than 0.2 but smaller than 0.7). Both sets of particles were subjected to 2D sorting, and the best particles of each subset were merged and sorted again to obtain a new particle stack of 884,341 particles. After 3D refinement, the final reconstruction had a slightly improved quality compared to the first processing attempt as judged by visual inspection of the map quality. However, it was also apparent that a significant number of 'good' particles (i.e. particles that were kept after 2D sorting with ISAC) were only selected in one of the two sorted particle stacks, indicating that about 30% of high-quality particles were lost during 2D sorting.

To further improve the quality of the sorted particle stack, we developed a procedure to combine multiple particle stacks from independent processing attempts. Briefly, we re-extracted box files for all stacks using the PIPE restacking procedure from SPHIRE, combined all box files for a given micrograph, and then removed all boxes which are in close proximity to another box in that image with a simple bash script. The thus cleaned set of boxes was used to re-extract a stack with unique, non-overlapping particles. The use of box files has the advantage that it is straightforward to inspect the success of the procedure, and even allows to identify duplicated particles with slightly different origin, for example when the original particle selections were updated in one of the processing attempts by recentering and particle re-extraction. By combining the two particle stacks from both independent processing attempts and removing all duplicate particles, we rescued about 220,000 "good" particles that were sorted out in one or the other processing attempt. The derived combined stack of 1,103,959 particles could thus be subjected to a final, more stringent 2D sorting round to obtain a final particle stack with 862,114 particles that, after 3D refinement, resulted in a 3.8 Å reconstruction with significantly improved quality compared to the individual refinement attempts.

3D clustering using SORT3D of the SPHIRE suite was performed with a 3D-focussed mask that includes BBS5, which was first apparent in a variability map of the full particle stack as calculated using 3DVARIABILITY from SPHIRE (*Figure 1—figure supplement 3*). 3D clustering separated the particles into four 3D clusters which were then subjected to local 3D refinement using MERIDIEN. One of the generated 3D reconstructions clearly indicated BBS5 density, resulting in a 4.3 Å resolution map of the core BBSome with BBS5 (*Figure 1—figure supplement 3*). An analysis of the 3D variability in this map indicated no remaining variability in the BBS5 focus region, while the variability in other regions of the complex remained similar as in the full particle stack (e.g. indicating some conformational heterogeneity around the 4α helical insert in the β-propeller of BBS1, around the platform domain of BBS9, at the tip of the hairpin within the 5α helical domain of BBS9, and at the unresolved C-terminal end of BBS4). All of these regions with enhanced variability indeed have a lower-than-average map resolution, indicating a high flexibility in these regions. Nevertheless, the 4α helical insert in the β-propeller of BBS1 in our structure was better defined than in the bovine BBSome structure (*Chou et al., 2019*), and allowed the building of a model.

## Model building

For building an atomic model of the BBSome core comples, we used the two available crystal structures of the β-propeller domains of *C. reinhardtii* BBS1 (*Mourão et al., 2014*) and of human BBS9 (*Knockenhauer and Schwartz, 2015*) as starting points for the assignments of BBS domains. For this, the human homology model to the *C. reinhardtii* BBS1 β-propeller was generated using HHpred (*Söding et al., 2005*), and the structure of a helical insert in the β-propeller (Pro127-Gln197) that is missing in the crystal structure was predicted using RaptorX (*Wang et al., 2018*). For the human BBS9 β-propeller, the pdb entry 4YD8 could be directly used as starting model after minor modifications (i.e. changing the selenomethionine residues to methionine and adding a few loops that were missing in the crystal structure). The large helical insert in BBS1 that is not present in BBS9 allowed an unambiguous placement of both structures into the density. BBS4 and BBS8 are the two subunits that are predicted to fold into TPR repeats, and from biochemical data it is apparent that BBS9

forms a stable binary subcomplex with BBS8 (26). Since in the cryo-EM map two superhelical arrangements of TPR motifs are visible and BBS9 forms no interactions with one of them, the correct placement of BBS9 and BBS1 also allowed an unambiguous assignment of BBS4 and BBS8. Initial models for BBS4 and BBS8 were generated by de novo structure prediction using RaptorX (*Wang et al., 2018*; *Wang et al., 2016*; *Wang et al., 2017a*; *Wang et al., 2017b*) followed by flexible fitting of fractions of the RaptorX structure predictions into consistent parts of the density using iMODFIT (*Lopéz-Blanco and Chacón, 2013*) and subsequent manual model correction in Coot (*Emsley et al., 2010*). The validity of the fit could be verified based on side chain consistency with the map as BBS4 and BBS8 are positioned in a central region of the complex with sufficiently high local map resolution. Consistent with the RaptorX prediction, we observed that a region of about 130 residues in BBS8 (Tyr53-Lys180) is only partially structured and forms an extended loop that winds through the center of the complex and back (*Figure 1—figure supplement 5A,B*).

The assignments of the C-terminal parts of BBS1 and BBS9 were less obvious as they are separated by partially flexible linkers from the N-terminal β-propeller domains. We could assign the GAE domains of BBS1 and BBS9 based on the clearly resolved connection of the BBS9 GAE domain to the C-terminal platform and α-helical domains that are missing in BBS1. Furthermore, the connection of the BBS1 GAE domain could be traced up to a small gap of about 20 residues to its corresponding N-terminal β-propeller, which would not allow an alternative assignment. The structures of all these domains were also predicted with RaptorX, flexibly fitted with IMODFIT and manually corrected with Coot as described above. The resolution of the GAE domains was sufficient to verify the validity of the model based on side-chain densities, and even in the lower resolved part of the platform and α helical domains of BBS9, multiple hallmark residue side chains allowed a clear validation of the overall fold, although some solvent-exposed loops are only weakly defined.

After placing BBS1, 4, 8 and 9, a prominent stretch of well resolved density remained unexplained that winds through the superhelices formed by BBS4 and BBS8 and connects to a short α-helix. With a total length of about 60 residues and no obvious covalent connection to one of the other subunits, we were confident that this stretch represents a major part of the 93 residue subunits BBS18, which is also consistent with biophysical data showing that BBS18 forms stable subcomplexes with BBS4 and with BBS4, 8 and 9 (*Klink et al., 2017*). With few initial indications how to assign the primary sequence to this linear, mostly unfolded domain, we utilized the following method to find the correct frame: We first built a 59 residue poly-alanine model, then assigned all 33 possible frames of BBS18 sequence to this model, and further refined each of them automatically using Phenix real-space refinement (*Adams et al., 2010*) (*Figure 1—figure supplement 4A–F*). Comparing the correlation of the map to the refined models, we observed a clearly separated best hit for one sequence assignment (*Figure 1—figure supplement 4A,F*) with no obvious discrepancies between map and model, while the other models showed significant deviations from map to model that could not be explained by noise, imperfect modeling or a lack of resolution (*Figure 1—figure supplement 4C,D, E,G*). The final model was further manually refined in Coot and corresponds to residues Val26-Gln80 of BBS18 (*Figure 1—figure supplement 4H*). As shown in *Figure 1—figure supplement 4I*, the final model forms favorable interactions with neighboring BBS subunits BBS4 and BBS8.

After placing all subunits except BBS5, missing residues that were visible in the density were placed and all subunits were further manually refined in Coot. Atom clashes were removed by energy minimization (with torsion and Ramachandran restraints turned on) using PHENIX real space refinement (*Adams et al., 2010*), followed by another round of manual refinement. Compared to the other subunits of the core BBSome, BBS5 is more loosely attached at the periphery of the complex, and could only be identified in a subpopulation of particles that was isolated by 3D clustering (*Figure 1—figure supplements 1* and *4*). The best 3D class containing BBS5 allowed a reconstruction with an average resolution of about 4.3 Å (*Figure 1—figure supplement 1D*, *Figure 1—figure supplement 3*). The two PH domains of BBS5 were modeled with RaptorX and were positioned into the derived BBS5 density using Chimera (UCSF) (*Pettersen et al., 2004*), followed by relaxation into the density in Coot (*Figure 1—figure supplement 1E*). An overlay of the maps derived from the full particle stack with the 3D cluster containing BBS5 allowed to position BBS5 relative to the higher resolved reconstruction from all particles in which BBS5 was not visible. BBS5 was then combined with the higher resolved full particle reconstruction to generate a combined molecular model of the human BBSome core complex.

## Visualization

Visualization, analysis and figure preparation was done with Chimera (UCSF) (*Pettersen et al., 2004*). Local resolution gradients within a map were estimated using LOCRES as implemented in the SPHIRE suite, and final densities were filtered according to the calculated local resolution unless stated otherwise. Resolution gradients were visualized by coloring the corresponding maps according to the local resolution in Chimera (*Figure 1—figure supplement 2D*). To visualize surface electrostatic potentials, the correct protonation state of the core BBSome was predicted using the H++ web server (http://biophysics.cs.vt.edu/H++ ; *Anandakrishnan et al., 2012*) and the hydrogenated model was colored according to its electrostatic potential in Chimera. 3D average and variability maps were calculated using 3DVARIABILITY of the SPHIRE package and visualized in Chimera. Angular distribution plots were generated using PIPE from SPHIRE. In addition to the binned 2-D class averages produced by ISAC that were used for the particle selection process, unbinned class averages showing high-resolution features were calculated using COMPUTE_ISAC_AVG from SPHIRE for visualization purposes.

## Acknowledgements

We thank Dr. Daniel Prumbaum for his support during EM data collection, the SPHIRE developer team for support in image processing and the whole Raunser lab for support and fruitful discussions. This work was funded by the Max Planck Society (to SR).

## Additional information

### Funding

| Funder | Author |
| --- | --- |
| Max-Planck-Gesellschaft | Stefan Raunser |

The funders had no role in study design, data collection and interpretation, or the decision to submit the work for publication.

### Author contributions

Björn Udo Klink, Data curation, Formal analysis, Visualization, Writing - original draft; Christos Gatsogiannis, Formal analysis; Oliver Hofnagel, Data curation; Alfred Wittinghofer, Conceptualization; Stefan Raunser, Conceptualization, Funding acquisition, Investigation, Project administration, Writing - review and editing

### Author ORCIDs

Björn Udo Klink (ID) https://orcid.org/0000-0002-0946-4456
Christos Gatsogiannis (ID) http://orcid.org/0000-0002-4922-4545
Alfred Wittinghofer (ID) http://orcid.org/0000-0002-5800-0236
Stefan Raunser (ID) https://orcid.org/0000-0001-9373-3016

### Decision letter and Author response

Decision letter https://doi.org/10.7554/eLife.53910.sa1
Author response https://doi.org/10.7554/eLife.53910.sa2

## Additional files

### Supplementary files

- Transparent reporting form

## Data availability

The electron density maps have been deposited to the EMDB under the accession codes EMD-10617 and EMD-10618. The final models of the BBSome were submitted to the Protein Data Bank under the accession codes 6XT9 (subunits BBS1,4,8,9,18) and 6XTB (subunits BBS1,4,5,8,9,18).

The following datasets were generated:

| Author(s) | Year | Dataset title | Dataset URL | Database and Identifier |
|---|---|---|---|---|
| Björn Udo Klink, Stefan Raunser, Christos Gatsogiannis | 2020 | Subunits BBS 1,4,8,9,18 of the human BBSome complex | https://www.ebi.ac.uk/pdbe/entry/emdb/EMD-10617 | Electron Microscopy Data Bank, EMD-10617 |
| Björn Udo Klink, Stefan Raunser, Christos Gatsogiannis | 2020 | The human core BBSome complex (BBS 1,4,5,8,9,18) | https://www.ebi.ac.uk/pdbe/entry/emdb/EMD-10618 | Electron Microscopy Data Bank, EMD-10618 |
| Klink BU, Raunser S, Christos Gatsogiannis | 2020 | Subunits BBS 1,4,8,9,18 of the human BBSome complex | https://www.rcsb.org/structure/6XT9 | RCSB Protein Data Bank, 6XT9 |
| Klink BU, Raunser S, Christos Gatsogiannis | 2020 | Subunit BBS 5 of the human core BBSome complex | https://www.rcsb.org/structure/6XTB | RCSB Protein Data Bank, 6XTB |

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
