## [Decision Letter]

**Acceptance summary:**

This manuscript describes the cryo-EM structure of a complex of 6 recombinantly expressed human BBSome proteins (BBS1, 4, 5, 8, 9 and 18), which form what the authors refer to as the 'core' complex (lacking BBS2 and 7). The maps are at sufficiently high resolution to enable the authors to build atomic models, including side chains, for the majority of the subunits, and map known Bardet-Biedel syndrome causing mutations.

This work comes in the context of the recently published 4.9 Å cryo-EM structure of the complete bovine BBSome from Chou et al., and a pre-print describing cryo-EM structures of the complete bovine BBSome with and without ARL6 at resolutions of 3.5 and 3.1 Å, respectively. There is significant overlap between these works, but also complementarity in the analyses presented. For example, in this study the authors analyze the charge distribution to propose a mode of interaction with transmembrane protein cargo. This study also involves a recombinant complex which will be important for future mutagenesis studies.

Importantly, despite lacking BBS2, 7, and ARL6, the BBSome core complex reported here exhibited an "open" conformation that can bind to the substrate, and the β-propeller of BBS1 was rotated compared to the "closed" conformation structure of Chou et al.

**Decision letter after peer review:**

Thank you for submitting your article "Structure of the human BBSome core complex in the open conformation" for consideration by *eLife*. Your article has been reviewed by three peer reviewers, and the evaluation has been overseen by a Reviewing Editor and John Kuriyan as the Senior Editor. The following individual involved in review of your submission has agreed to reveal their identity: Masahide Kikkawa (Reviewer #1).

The reviewers have discussed the reviews with one another and the Reviewing Editor has drafted this decision to help you prepare a revised submission.

Essential revisions:

1) EM processing and model building is described in great detail (subsection “Model building”) but it is unclear to me how the models were refined. Refinement and validation statistics are given in Table 1, which suggest that the models were indeed refined. Was real-space refinement in Coot used for the entire BBSome core structure and Phenix only used for energy minimization or was real space refinement also carried out in Phenix? If so, which restraints/constraints were used? It might be useful to include a refinement paragraph in the Materials and methods section describing the procedures used. It is also not clear to me if the last column in Table 1 reports refinement/validation statistics for the full core complex (BBS1,4,5,8,9,18) or only the BBS5 subunit given the BBS5 label for this column. The Ramachandran scores in Table 1 (in particularly for the column with the BBS5 label) are highly unusual with 0% outliers but only 80.5% in the favored regions. The 0% outliers suggest that Ramachandran restraints were used in refinement. 80% in the favored region is a very low number and strongly suggests that there are severe problems with at least part of the structure. The authors should have a careful look at regions with a large percentage of residues not falling into the favored regions and see if there are issues to be fixed.

2) Figure 1—figure supplement 2G and I: The fit of the model to the density is not super convincing. The lower part of the helix in insert G and a loop region of the GAE domain in insert I appears to be out of density. The authors may need to put some more effort into improving the initial models to obtain a structure that fits the experimentally obtained maps better. This may also help to improve the poor/unusual Ramachandran plot scores discussed in comment 1 above.

3) Figure 1—figure supplement 4: sequence assignment of BBS18. I am not convinced that the authors have assigned the entire sequence of BBS18 correctly. The authors judge the frame based purely on correlation with the map but do not appear to make use of the chemical environment of BBS18 in deciding on the correct frame. In the PDB file associated with the submitted manuscript, there are several clashes between BBS18 and sidechains from neighboring subunits and the chemistry of non-covalent interactions makes little sense in some places. For example, the side-chain of BBS18 M29 clashes with M417 of chain D; BBS18 E32 sidechain and R31 carbonyl clashes with the hydrophobic sidechain of Leu381 of chain D. This strongly suggests that the sequence of BBS18 is at least partly incorrectly assigned in the current structure. This appears to be mostly a problem for the N-terminal part of BBS18 whereas the C-terminal part (residues 37-80) makes chemical contacts that are much more plausible. Could the authors please correct the structure and provide a figure showing the interaction of BBS18 including sidechains with the neighboring BBS subunits so that the reader can judge for herself? Figure 3B does include some sidechains of BBS18 but not of interacting BBS subunits so it is impossible to judge if the frame chosen for BBS18 makes chemical sense with respect to non-covalent interactions. In addition, for Figure 1—figure supplement 4, instead of showing the density and fit of two 'bad models' (frame 2 and 5, panels C and D), it may be more informative to show only one 'bad model' and then also the second best fit (frame 33). Also, there are a total of 11 frames with correlation coefficients very similar (CCs of 0.79-0.81) to frame 22 that was chosen as the correct frame by the authors. The figure legend states that frame 22 is in full agreement with the observed density. This is clearly an overstatement. Several side-chains have missing density and one F has additional density at the tip of the aromatic ring. I suggest rewriting the sentence to better reflect what is shown in the figure. I fully appreciate that assigning the sequence to a mostly unstructured protein in a 3.8Å resolution map is a difficult task and one could take the more conservative approach of simply leaving part of BBS18 modelled as poly-alanine.

4) Open conformation: The authors describe their structure as the 'open conformation' in the title. Although it is plausible that the core complex resembles the open state of the intact BBSome, it could also be a non-physiological conformation adopted in the absence of BBS2 and BBS7. Therefore, 'open conformation' should be removed from the title, and the arguments and caveats for the core complex representing the open state should be elaborated in the main text.

5) Gels showing the purified 4mer and 6mer sub-complexes would be helpful.

6) The structure of BBS5 is at medium resolution, which does not seem sufficient to confidently build side chains. Please include examples of density to justify keeping side chains in the model or adjust model accordingly.

---

## [Author Response]

Essential revisions:1) EM processing and model building is described in great detail (subsection “Model building”) but it is unclear to me how the models were refined. Refinement and validation statistics are given in Table 1, which suggest that the models were indeed refined. Was real-space refinement in Coot used for the entire BBSome core structure and Phenix only used for energy minimization or was real space refinement also carried out in Phenix? If so, which restraints/constraints were used? It might be useful to include a refinement paragraph in the Materials and methods section describing the procedures used.

We indeed performed real-space refinement on the entire BBSome structure in Coot, and used Phenix only for energy minimization (using standard settings, with Ramachandran and Torsion restraints). We expanded the description of refinement in the revised manuscript (subsection “Model building”).

It is also not clear to me if the last column in Table 1 reports refinement/validation statistics for the full core complex (BBS1,4,5,8,9,18) or only the BBS5 subunit given the BBS5 label for this column.

The last column in Table 1 was only for subunit BBS5. Since we followed the suggestion to cleave back BBS5 to poly-alanine (see below), the second column became obsolete and we changed Table 1 to a single column, with BBS5-related numbers in brackets. This should remove the confusion concerning the table.

The Ramachandran scores in Table 1 (in particularly for the column with the BBS5 label) are highly unusual with 0% outliers but only 80.5% in the favored regions. The 0% outliers suggest that Ramachandran restraints were used in refinement. 80% in the favored region is a very low number and strongly suggests that there are severe problems with at least part of the structure. The authors should have a careful look at regions with a large percentage of residues not falling into the favored regions and see if there are issues to be fixed.

The low percentage of residues in the favored regions was only for the BBS5 subunit, while the rest of the structure has much better statistics. As we described in the Materials and methods section, we only positioned the RaptorX model of BBS5 into the density using Chimera and relaxed it into the density in Coot, but the quality of the map did not justify more detailed refinements, which explains the bad statistics. In the revised manuscript, we followed the suggestion to remove the sidechains of BBS5 to avoid overinterpretation (see below) and changed the BBS5 model to poly-alanine. With this change, the refinement statistics of the BBS5 subunit is not meaningful anymore. We removed the second column in Table 1 and added the relevant information for the BBS5-containing 3D cluster in brackets to the first column.

We also updated the first column since our refinement statistics slightly changed after changing an unstructured loop in BBS1 (residues 410-425) and the last two helices of BBS4 (see below) to poly-alanine.

2) Figure 1—figure supplement 2G and I: The fit of the model to the density is not super convincing. The lower part of the helix in insert G and a loop region of the GAE domain in insert I appears to be out of density. The authors may need to put some more effort into improving the initial models to obtain a structure that fits the experimentally obtained maps better. This may also help to improve the poor/unusual Ramachandran plot scores discussed in comment 1 above.

We updated Figure 1—figure supplement 2 with the optimized model and densities in which the fit to the map can be seen better. For this we also changed the threshold of the map to better visualize sidechain densities. For the GAE domain in insert I, we tried to find a loop conformation that fits the map better, but did not find another conformation of that loop that does not clash with other regions of the protein or is in an unreasonably strained conformation. We therefore conclude that minor discrepancies between map and model in the loop might be due to some noise in the map, which in this region is very close to the exterior of the complex.

3) Figure 1—figure supplement 4: sequence assignment of BBS18. I am not convinced that the authors have assigned the entire sequence of BBS18 correctly. The authors judge the frame based purely on correlation with the map but do not appear to make use of the chemical environment of BBS18 in deciding on the correct frame. In the PDB file associated with the submitted manuscript, there are several clashes between BBS18 and sidechains from neighboring subunits and the chemistry of non-covalent interactions makes little sense in some places. For example, the side-chain of BBS18 M29 clashes with M417 of chain D; BBS18 E32 sidechain and R31 carbonyl clashes with the hydrophobic sidechain of Leu381 of chain D. This strongly suggests that the sequence of BBS18 is at least partly incorrectly assigned in the current structure. This appears to be mostly a problem for the N-terminal part of BBS18 whereas the C-terminal part (residues 37-80) makes chemical contacts that are much more plausible.

We agree that the placement of the mostly unstructured BBS18 subunit at 3.8 Å is a difficult task that bears the potential of a mis-assignment of the correct frame. The N-terminus of BBS18 is less well defined as the C-terminal part, which leads to some ambiguities in the placement of sidechains in the last loosely attached residues. The described clashes, however less an issue in the BBS18 chain itself than in the interacting C-terminal helices of BBS4 (residues 386-423), which are poorly resolved and indeed do not justify the placement of sidechains. We therefore cleaved these residues back to poly-alanine and rebuilt the BS18 N-terminus without these potentially misleading constraints. A better fit of the BBS18 N-terminus to the map further supports the validity of our frame assignment.

Could the authors please correct the structure and provide a figure showing the interaction of BBS18 including sidechains with the neighboring BBS subunits so that the reader can judge for herself? Figure 3B does include some sidechains of BBS18 but not of interacting BBS subunits so it is impossible to judge if the frame chosen for BBS18 makes chemical sense with respect to non-covalent interactions. In addition, for Figure 1—figure supplement 4, instead of showing the density and fit of two 'bad models' (frame 2 and 5, panels C and D), it may be more informative to show only one 'bad model' and then also the second best fit (frame 33). Also, there are a total of 11 frames with correlation coefficients very similar (CCs of 0.79-0.81) to frame 22 that was chosen as the correct frame by the authors.

We agree that it is more informative to also show the second best fit (frame #33). We updated Figure 1—figure supplement 4 accordingly. There are several discrepancies when fitting frame #33 into the map, but even more importantly, this assignment (like many of the other potential frame assignments) would position several charged residues into a highly unfavorable chemical environment, while the assignment for frame #22 seems favorable. To be able to judge the validity of the fit, we included a panel showing a comparison between the second-best and the best frame (Figure 1—figure supplement 4G), and three more panels showing the interactions of BBS18 with neighboring BBS subunits.

The figure legend states that frame 22 is in full agreement with the observed density. This is clearly an overstatement. Several side-chains have missing density and one F has additional density at the tip of the aromatic ring. I suggest rewriting the sentence to better reflect what is shown in the figure. I fully appreciate that assigning the sequence to a mostly unstructured protein in a 3.8Å resolution map is a difficult task and one could take the more conservative approach of simply leaving part of BBS18 modelled as poly-alanine.

We have changed our phrasing concerning the map/model agreement of BBS18 as suggested. By “Full agreement with the observed density” we wanted to emphasize that the discrepancies between map and model are within the expected range at 3.8 Å resolution. This means that clearly resolved sidechain density should be explained by an appropriate residue, while negatively charged sidechains or residues with the potential to “wobble around” are not unexpected to have missing density at this resolution.

4) Open conformation: The authors describe their structure as the 'open conformation' in the title. Although it is plausible that the core complex resembles the open state of the intact BBSome, it could also be a non-physiological conformation adopted in the absence of BBS2 and BBS7. Therefore, 'open conformation' should be removed from the title, and the arguments and caveats for the core complex representing the open state should be elaborated in the main text.

We removed “open conformation “from the title and Introduction. In addition, we added a clarifying sentence in the Results and Discussion section.

5) Gels showing the purified 4mer and 6mer sub-complexes would be helpful.

Since we did not change the purification strategy from our 2017 paper (Klink et al., 2017), and the protein-lipid overlay assays and initial cryo-EM analysis was actually performed with the protein samples that are shown in that paper, we felt that representing this data is redundant. We added a precise reference to where this gel can be found in the Materials and methods section.

6) The structure of BBS5 is at medium resolution, which does not seem sufficient to confidently build side chains. Please include examples of density to justify keeping side chains in the model or adjust model accordingly.

Our initial model for BBS5 was generated by de novo structure prediction using RaptorX, followed by relaxation of the model to the map in Coot. We agree that there are too many poorly resolved parts in the BBS5 map that cannot unambiguously be refined. We therefore changed the model of BBS5 to a poly-alanine model.